# STRUCTURAL CODE REPRESENTATION LEARNING FOR AUTO-VECTORIZATION

## ABSTRACT

The single instruction multiple data (SIMD) capability in modern processors is critical to improving the performance of current compute-intensive programs. SIMD allows architectures to exploit the natural data parallelism that exists in a wide-range of real applications (e.g., games, signal processing, etc) by executing a single instruction on multiple data items simultaneously. Modern compilers use vectorization techniques to exploit the SIMD capability, by detecting data parallelism in scalar source code and transforming a group of scalar instructions into vector-based instructions. In this work, we focus on one of the most common vectorization techniques called *loop-based vectorization*, which targets loops and optimize their performance by grouping multiple occurrences of the same operation across loop iterations into single SIMD instructions. This is achieved by setting two key parameters: (1) the vectorization factor (VF), and (2) the interleaving factor (IF). Unfortunately, vectorizing loop computations effectively is a key challenging problem for both programmers and compilers due to the large search space. For example, manual vectorization of each loop puts a huge burden on the programmer, is more error-prone, and/or requires expert knowledge of both the software and the architecture. Alternatively, current compilers use fixed-cost models based on expert heuristics to make automatic vectorization decisions. However, these models often ignore the data dependencies, as well as the underlying computation graph. In this paper, we propose a data-driven graph-based learning framework for automatic vectorization, called *autograph*, which takes an input program, extracts the loops, then learns a structured representation to automatically predict the correct VF/IF factors. Our proposed framework utilizes deep reinforcement learning to learn an optimal policy (observations to actions) from an intelligent agent in a SIMD environment, and automatically injects the predicted vectorization pragmas into the input program. We conducted an extensive evaluation on multiple benchmark datasets and comparisons with state-of-the-art baselines. Our results show that for Polybench, autograph achieves on average 2.47x performance improvement for Polybench compared to neurovectorizer and 3.61x compared to the baseline.

## 1 INTRODUCTION

The single instruction multiple data (SIMD) mechanisms have been widely incorporated in modern processors such as gaming machines, massively parallel supercomputers, as well as general-purpose processors (Nuzman et al., 2006; Bachega et al., 2004; Peleg & Weiser, 1996). These mechanisms allow architectures to exploit the natural parallelism that exists in software for real-world applications (e.g., games, signal processing, etc.), by simultaneously executing the same instruction on multiple elements of the input data. Modern compilers use vectorization techniques to exploit the SIMD capability of these architectures. Vectorization techniques allow the compiler to reveal the data parallelism in the scalar source code and converts the code from a scalar implementation to the corresponding functionally-correct vector implementation.This allows portions of the code to run on the processor's high-throughput SIMD units, without any additional effort from the programmer (Porpodas et al., 2018). With the SIMD architecture, such operations can run in fewer cycles while using less energy to boost performance in applications with vector computations.

Vectorization can be classified into two major methods: (*i*) the loop vectorizer, which operates on loops, and (*ii*) the superword-level parallelism (SLP) vectorizer (Porpodas, 2017; Mendis et al.,

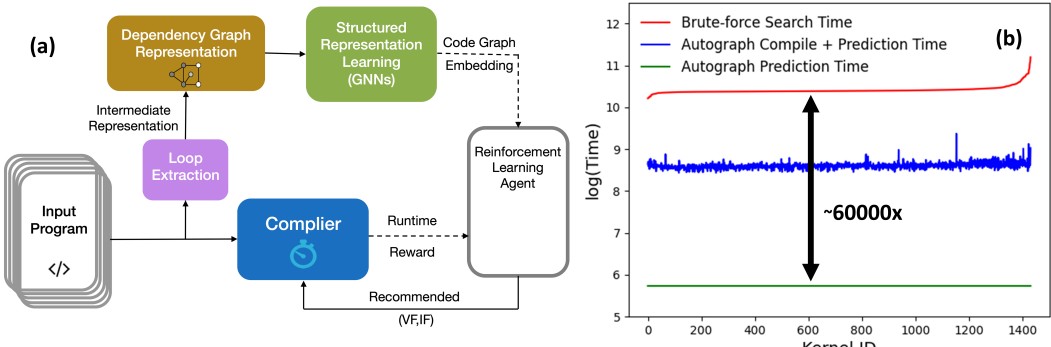

Figure 1: (a) Overview of the proposed autograph framework. Autograph extracts loops from source code in the intermediate representation format (LLVM IR). Then using compiler dependency analysis, autograph constructs dependency graphs to capture the flow of information in the code, as well as the semantics of the code by using the full text features of the instructions. Autograph then learns a structured representation using an inductive GNN approach. Finally, autograph expolits deep reinforcement to learn a mapping from embeddings to vectorization factors. (b) Comparison between brute-force and prediction-based autograph. For each kernel, the brute-force search time is measured by summing up the compile times and execution times shown in a $log_{10}$ scale of nanoseconds with different VFs and IFs. Although the brute-force can eventually find the best vectorization parameters, it is $\sim$60 000x slower compared to autograph that only predicts the parameter without exhaustively searching the space.

2019), which operates on straight-line code. Loops in repetitive tasks are commonly used in modern programs to save time and code size. Therefore, in this work, we focus on the loop vectorizer. One of the key challenges is to define the vectorization factor (VF) and the interleaving factor (IF) (Nuzman et al., 2006). The VF determines how many instructions to pack together from different iterations of the loop. The IF determines the stride of the memory accesses of the packed instructions (Haj-Ali et al., 2020). Hence, the goal of loop vectorization is to search for the optimal VF and IF, given a program. As shown in Figure 1(b), even though brute-force search has the ability to find the optimal/best vectorization parameter, the compiler cannot support the exhaustive search due to its time-consuming process. The brute-force method exhaustively searches the vectorization parameter for each loop in a kernel while taking a significant amount of time. For a kernel with $N$ loops with $M$ interleaving factors (IFs) and $K$ vectorization factors (VFs), the search space is $O(NMK)$.

Since manual vectorization is error-prone and difficult to maintain, modern compilers such as LLVM use auto-vectorization techniques that rely on linear and constant-cost models to predict the vectorization factors (Tian et al., 2016; Trifunovic et al., 2009). However, these cost models don't consider the computation graph or loop dependencies, thus, they are not capable of capturing the natural structural dependencies and semantics of the wide-range of software programs that exist in today's real-world applications. Machine learning has been proposed to improve these cost models (Stock et al., 2012; Wang & O'Boyle, 2018) by extracting hand-engineered features from assembly code and using supervised learning techniques to predict the vectorization factors. However, these methods are still incapable of automatically learning a representation that can capture the computation graph, as well as the dependencies of input codes. As we will show in evaluation, by accounting for structural dependencies we improve performance by 1.26x.

In this work, we propose a framework that learns a representation that is capable of reasoning about the flow of information and the semantics in the code, while capturing the structural dependencies in the computation graph. More specifically, in Figure 1(a), we propose an end-to-end graph-based deep learning framework for compiler auto-vectorization, called *autograph*, that takes an input code, extracts the loops, then learns a structured representation to automatically predict the correct VF/IF factors. Autograph utilizes deep reinforcement learning to learn an optimal policy (graph embeddings to VF/IF pairs) from an intelligent agent in a SIMD environment, and automatically injects the predicted vectorization pragmas into the input program to achieve better performance. We conducted an extensive evaluation on multiple benchmarks and comparisons with state-of-the-art baselines. Our experimental results show that for Polybench, autograph achieves on average 2.47x performance improvement for Polybench compared to neurovectorizer and 3.61x compared to the baseline.

We summarize the main contributions as follows:

- We propose an end-to-end graph-based deep learning framework for compiler auto-vectorization, *autograph*, that is capable of reasoning about the flow of information and the semantics in the code, while capturing the structural dependencies in the computation graph.
- We develop a deep reinforcement learning approach within the loop vectorizer capable of learning the optimal vectorization factors of a complex program.
- The proposed autograph framework is general and flexible to accommodate different representations of code to predict the vectorization factors.

## 2 RELATED WORK

Automatic vectorization has been widely studied for improving application performance on SIMD architectures (Nuzman et al., 2006; Tian et al., 2016). Some of the existing work in auto-vectorization focuses on designing cost models for calculating expected execution time of vectorized loops (Trifunovic et al., 2009; Nuzman et al., 2011). For example, Trifunovic et al. (2009) develops a cost model to compare the performance of various vectorization alternatives and their interactions with other loop optimizations. Other work relies on heuristics for automatic vectorization, such as the vector-instruction aware and heuristic-guided search system proposed in (McFarlin et al., 2011) for automatic vectorization of fast Fourier transforms.

In recent years, some research has also explored machine learning strategies for compiler optimization (Leather & Cummins, 2020; Trofin et al., 2021; Wang & O'Boyle, 2018). For instance, the SuperGraph-SLP (Porpodas, 2017) and VW-SLP (Porpodas et al., 2018) propose improved vectorization algorithms based on Superword-Level Parallelism (SLP). The SuperGraph-SLP operates on a larger region to enable it to vectorize code that was previously unreachable. The VW-SLP varies the vector width at the granularity of the instruction level. This allows the algorithm to better adjust the code in SIMD and provide more parallelism. The MLGO framework (Trofin et al., 2021) introduces machine learning methods such as policy gradient and evolution strategies to LLVM compiler optimizations. Mammadli et al. (2020) employs a deep reinforcement learning (DRL) method to the phase ordering in one of the compiler optimizations. Neuro-vectorizer (Haj-Ali et al., 2020) develops a DRL approach for loop-level auto-vectorization. However, (Wang & O'Boyle, 2018) only provides a general survey of how to do feature engineering and provide optimization for compilers using machine learning. In addition, (Mendis et al., 2019) focuses on SLP vectorization whereas the focus of this work is loop vectorization. Therefore, both of them are not proper baselines.

In contrast to neuro-vectorizer, autograph is the first to propose a structured learning framework for loop auto-vectorization, which converts codes that contain loops to graphs and learns structured representations from code graphs. This structured representation can better capture the dependencies among instructions compared to previous work that only relies on the code text features. Autograph can be used across different architectures with different SIMD capabilities. The only requirements of autograph are (1) the ability to run applications on such architecture so we can collect the true label; and (2) compiler support to compile code with VF and IF into SIMD instructions.

## 3 PROPOSED FRAMEWORK

In this work, we propose an end-to-end graph-based deep learning framework, called *autograph*, to predict vectorization factors. The proposed autograph framework is shown in Figure 1(a). Target programs are translated to obtain low-level virtual machine (LLVM) intermediate representation (IR) instructions of extracted loops to be vectorized. The IR instructions are then analyzed via compiler dependency analysis in terms of data, control, and function calls to construct dependency graphs. The graphs are then fed into a structured representation learning module that includes graph unsupervised learning and supervised learning to generate hidden embeddings. Finally, these embeddings are fed to the DRL agent to predict the vectorization factors VF and IF. The agent automatically injects vectorization pragmas, e.g., `#pragma clang loop vectorize_width(32) interleave_count(16).` The agent uses the *clang* compiler to generate the executable of the program and runs it to gather the runtime, which is further used to calculate a reward for the RL agent. During inference / testing, the well trained agent can be exploited to predict VF/IF pairs and find

the corresponding runtime information. The autograph framework injects pragmas only as hints to help the compiler to generate the vectorized assembly instructions of loops such as Advanced Vector Extensions (AVX) in Intel architecture.

**Graph representation of code.** Code captures computations and communications within and between different high-level functions. Therefore, it is of great importance to understand the heterogeneous structural organizations and dynamics of programs such as data flow from one instruction to another. However, traditional data-flow and control-flow graphs do not reveal the relationship between programs and underlying resources. Many recent graph representations of code Xiao et al. (2019; 2021; 2017) can identify the spatio-temporal information flow by analyzing data dependencies between LLVM IR instructions collected at run-time. Moreover, PrograML Cummins et al. (2021) is a graph representation for programs as input to a machine learning model. The graph incorporates data-flow, control-flow, and call-flow that closely match the data structures used traditionally in inter-procedural data flow analysis. In our implementation, we use PrograML to generate code graphs.

**Inductive representational graph unsupervised learning.** From the code graphs, the autograph framework learns a representation that can reason about the information flow and the semantics in the code, while capturing the structural dependencies in the computation graph. Therefore, we propose inductive graph representation learning to learn embeddings of the nodes in an unsupervised way to understand semantics and structures of the code. The objective is formally formulated as follows.

*Given* a graph $G(V, E)$, *learn* an embedding of the nodes from the graph topological structures and the node attributes such as LLVM IR, without using any node labels.

The node embeddings are learned in a classification task: given a set of positive node pairs generated from random walks on the target graph and negative node pairs that are randomly selected from the graph, learn a binary classifier that predicts whether the existing node pairs in the graph are likely to appear in a random walk. In this way, the classifier is able to learn a mapping from each graph with node attributes and structural dependencies in the code into an embedding.

The architecture of the classifier is defined as follows. Node pairs together with initial features are passed into a layer that (1) uses a mean aggregator that aggregates the representations of each node in its immediate neighborhood into a vector, (2) concatenates the node's current representation, and (3) is fed through a fully connected layer with the ReLU activation function followed by the L2 normalization. Therefore, the classifier generates the embedding for each node in a graph. The general math operation of the forward pass of this model can be described as follows.

$$h_{N(i)}^{(l+1)} = mean\_aggregate(h_j^l, \forall j \in N(i)) \tag{1}$$

$$h_i^{(l+1)} = \sigma(W \times concat(h_i^l, h_{N(i)}^{(l+1)} + b)) \tag{2}$$

$$h_i^{(l+1)} = L2\_norm(h_i^l) \tag{3}$$

where $h$ encodes the hidden representations of the nodes; $N$ is a neighborhood function: $v \to 2^V$, $mean\_aggregate$ is a differentiable mean aggregator function; $\sigma$ is a fully connected layer followed by the ReLU activation function; $W$ is the weight matrix; $L2\_norm$ is the L2 normalization function.

Next, we use the dot product of the incident nodes' representations for each edge to calculate the edge score. Once the node-level embedding and edge score are calculated, we use binary cross entropy loss with the help of scores as the loss function as follows.

$$s_{u \sim v} = h_u^{(l+1)} \cdot h_v^{(l+1)} \tag{4}$$

$$L = -\sum_{u \sim v \in N} s_{u \sim v}(y_{u \sim v} log(\hat{y}_{u \sim v}) + (1 - y_{u \sim v}) log(1 - \hat{y}_{u \sim v})) \tag{5}$$

where nodes $u$ and $v$ are selected from node pairs $N$; $y_{u \sim v}$ and $\hat{y}_{u \sim v}$ are the predicted and true link labels between $u$ and $v$, respectively.

*Feature Representation.* The initial features used in the unsupervised graph model are directly collected from the full text feature in each node of a graph. The graphs are generated from PrograML

to capture the information flow between instructions. Therefore, associated with each node is one LLVM IR instruction. We tokenize all of the instructions to create a vocabulary that contains high-level structures, types, constants, and syntax. Next, for each full text in a node, if a token exists in this vocabulary, we mark the corresponding index as 1, and 0 otherwise. Therefore, by adopting this approach, the size of the node embedding is the same as the size of the vocabulary.

**Refined latent embedding supervised learning.** One drawback in the graph representation of codes is that two graphs could be identical but the vectorization parameters are totally different, due to different number of iterations in the loops. This makes it difficult to learn optimal VF/IF pairs for performance improvement.

Therefore, in order to mitigate this issue, based on the feature embeddings learned from the unsupervised model, we further refine them into the supervised model to provide better accuracy and performance improvement. The architecture of the supervised model is one layer of graph neural network (GNN), connected with one fully connected feed-forward neural network (FCNN), shown in Figure 2. In order to take into consideration the number of iterations in each for-loop, we insert them into the FCNN. Therefore, the hidden embedding size going into the FCNN is $N_{GNN} + N'$, where $N_{GNN}$ is the embedding size for the GNN and $N'$ is the number of loops in each file of a benchmark.

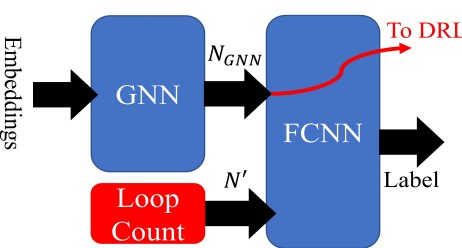

Figure 2: The architecture of the supervised model.

**Deep reinforcement learning.** So far, autograph generates structural embeddings for code graphs via graph unsupervised learning and further refines them in terms of better accuracy and performance improvement over O3 via graph supervised learning. Next, autograph learns the optimal mapping from the embeddings to the VF and IF pairs in an end-to-end fashion with reinforcement learning (RL). The rationale of employing the proposed end-to-end RL framework is that (1) it does not require labels to train the agent to prevent the time-consuming exhaustive search; (2) compared to supervised learning that optimizes the accuracy, it relies on the reward function that involves the normalized execution time to improve the system-level speedup; (3) it is hardware-agnostic, meaning that while switching to another hardware platform, this RL framework does not require the exhaustive search to collect new labels before training compared to supervised learning. In RL, an agent continually interacts with an environment to learn and maximize the reward. Unlike supervised learning, RL can be tuned to co-optimize multiple objectives and be more efficient in terms of samples as it does not require a label.

Therefore, in this section, we propose a deep RL (DRL) based mapper. We seek to overcome the drawbacks of the brute force, random search, and supervised learning to improve the accuracy and the performance of the vectorized code over O3 at run-time. We represent the process of mapping the embeddings to the VF and IF pairs as a Markov decision process (MDP) and apply DRL to learn the optimal mapping given an embedding as a state and a reward function for the quality of a mapping. We represent the vectorized kernels running on a specific hardware platform as an environment, and VF/IF pairs as a set of actions to maximize the reward function.

*Markov decision process (MDP).* MDP is a discrete-time stochastic control process that can solve memoryless RL problems in decision making. At each time step $t$, the process is in some state $s_t$ and an action $a_t$ is chosen to find the next state $s_{t+1}$ with an unknown probability $p(s_{t+1}|s_t, a_t)$. An agent $A(a_t|s_t; \theta)$ selects a sequence of actions followed by an optimal policy $\pi^*$ via learned parameters $\theta$. The agent activates the action $a_t$ under the state $s_t$ in the environment and the reward $r(s_t, a_t)$ is returned back to the agent.

We consider the selection of vectorization parameters VF / IF as an MDP. The state space consists of hidden feature embeddings learned from the supervised model. The action space consists of different discrete vectorization parameters. The agent's task, then, is to pick VF and IF for a given embedding from the code at given $s_t$ to maximize a reward function under the environment.

*Reward function.* The reward function is used to evaluate and learn the optimal policy. We design an immediate reward below for the current state $s_t$ under the action $a_t$.

$$r(s_t, a_t) = \frac{(t_{baseline} - t(s_t, a_t))}{t_{baseline}} \tag{6}$$

where $t_{baseline}$ is the execution time of a file running with the $-\mathtt{O3}$ flag in the LLVM workflow; $t_{s_t, a_t}$ is the execution time for the current VF / IF pair.

We calculate the difference of runtimes between the current vectorization $a_t$ for the kernel $s_t$ and the baseline running with the compiler flag O3 in LLVM. Next, it is normalized by the baseline runtime so that the reward function is robust to variations in the programs' execution times.

*Agent: fully connected neural network (FCNN).* The agent is used to select an action $a_t$ (VF/IF pair) from the state $s_t$ (embeddings) to find the optimal policy that maximizes the cumulative reward $R = \sum_{t=1}^{T} \gamma^t r(a_t, s_t)$ where $\gamma \in [0, 1]$ is a decay factor. The inputs to the agent are feature embeddings extracted from graphs via supervised models. Q values are used to represent the maximum cumulative reward the agent can obtain by taking the action $a_t$ in the state $s_t$. Then, the optimal value $Q^*(s_t, a_t)$ can be calculated by Bellman equation Sutton & Barto (2018) as follows.

$$Q^*(s_t, a_t) = E[r_{t+1} + \gamma \max_{a_{t+1}} Q * (s_{t+1}, a_{t+1} | s_t, a_t)] \tag{7}$$

We then follow Q-learning to update Q values.

$$Q_{t+1}(s_t, a_t) \leftarrow (1 - \alpha) Q_t(s_t, a_t) + \alpha[r(s_t, a_t) + \gamma \max_{a_{t+1}} Q_t(s_{t+1}, a_{t+1})] \tag{8}$$

where $\alpha \in [0, 1]$ is the learning rate.

*Environment: vectorized code running on hardware.* The environment interacts with the agent to receive possible actions (VF/IF pairs). Once an action is obtained, the environment injects the vectorization pragma into the code, compiles it into the executable, and runs it to get the runtime information. Next, the environment provides an immediate reward (eq. 6). This reward is returned back to the agent to help make better decisions next time.

## 4 EVALUATION

In this section, we discuss the hardware configurations, baselines, and datasets, along with experimental results to investigate the effectiveness of the proposed autograph framework.

### 4.1 EXPERIMENTAL SETUP

**Configurations.** The unsupervised learning, supervised learning, and DRL are performed in Ubuntu 20.04.3 LTS with 64 CPU threads and 2 GPUs. The CPU is an Intel(R) Xeon(R) Gold 5218 CPU at 2.30 GHz. One GPU is an NVIDIA TESLA V100, the other is an NVIDIA TITAN RTX TU102.

**Baselines.** The autograph framework incorporates GraphSAGE (Hamilton et al., 2017) to extract feature embeddings in an unsupervised fashion, GCN (Kipf & Welling, 2016) to further refine the embeddings to improve accuracy and performance, and DRL. DRL is implemented with RLlib (Liang et al., 2017) and Tune (Liaw et al., 2018), built on top of Ray (Moritz et al., 2018), open-source libraries for RL that offer high scalability, hyper-parameter tuning and numerous application interfaces. In the GraphSAGE setup, the number of layers is 1 and learning rate is 0.001 with the dropout rate of 0.1. In the GCN setup, the number of layers is 2 and learning rate is 0.001. In the DRL setup, the learning rate is 0.0001 with 800,000 steps and 15,000 training batch size. The sizes of hidden layers are both 256. The neuro-vectorizer framework incorporates code2vec (Alon et al., 2019) and DRL. The open-source code code2vec is modified to work with the RL agent implementation (Haj-Ali et al., 2020). The DRL hyperparameters are the same as autograph. We run a brute-force search on all of the datasets to find the optimal run-times and the best vectorization factors VF and IF, and use them as labels for supervised models. We also implement two supervised models, one with code2vec and FCNN and another one with GCN and FCNN. The batch size is 64. Both FCNNs have 2 layers with 0.001 learning rate. We run all of the models 5 times and report mean and standard deviation.

Table 1: Impact of different pipelines.

| Pipelines | Accuracy | Speedup |
|---|---|---|
| GraphSAGE-GCN | 18.45%±1.24% | 1.47± 0.14 |
| GCN | 13.22%±1.16% | 1.23±0.21 |
| GraphSAGE-Pooling-FCNN | 12.67%±1.45% | 1.17±0.15 |

Table 2: Impact of different GNN architectures.

| GNN | Accuracy | Speedup |
|---|---|---|
| GCN | 18.45%±1.24% | 1.47± 0.14 |
| GAT | 15.30%±1.17% | 1.32 ± 0.11 |
| GGNN | 19.12%±0.93% | 1.52± 0.12 |

**Datasets.** We use LORE suite of kernels (Chen et al., 2017) to train and test DRL. Specifically, we use the SPEC 2006 (Henning, 2006), NPB (Bailey et al., 1995), and Polybench (Pouchet, 2012) kernels as testing, and the rest of benchmarks as training. In addition, we transfer the learned model in DRL to further test the GCC (Haj-Ali et al., 2020) and Mibench (Guthaus et al., 2001) benchmarks. GCC contains a set of loops that was created to test the auto-vectorization capabilities of GCC. We parameterized the loops in terms of a number of arrays, arithmetic operations and data sizes to increase the number of kernels to 744. Mibench is a set of representative embedded benchmarks such as telecommunication, networking, security, office, and automation; we use the MiBench subset selected in the neuro-vectorizer paper.

## 4.2 Framework Comparison

**Training loss and reward mean.** We validate how well autograph is trained in terms of training loss and reward mean for LORE with different batch sizes and fully-connected neural networks (FCNN). As we can see from Figure 3, if the number of timesteps is small (< 100,000), the agent in DRL is unable to explore most of the high-dimensional state space and thus fails to converge to an optimal policy. Therefore, training loss is high and reward mean is small. But as we gradually increase the number of timesteps, the agent can explore the region and find a better policy. Note that there is also a diminishing return when we increase the number of timesteps further beyond a point where the training loss and reward almost converge to a number. Therefore, in the following evaluation, we set the number of timesteps to $800,000$ to ensure that the agent is well trained, leading to small training loss and high reward mean.

**Ablation - different pipelines.** In this part, we investigate the impact of graph unsupervised learning (to understand the structures of graphs) and graph supervised learning (to further refine the features). As shown in Table 1, with graph unsupervised learning such as Graph-SAGE, GraphSAGE-GCN can provide 1.40x higher in terms of accuracy and 1.20x in terms of speedup, compared to only GCN. With the graph supervised learning model such as GCN, GraphSAGE-GCN can provide 1.46x higher in terms of accuracy and 1.26x in terms of speedup, compared to GraphSAGE-Pooling-FCNN.

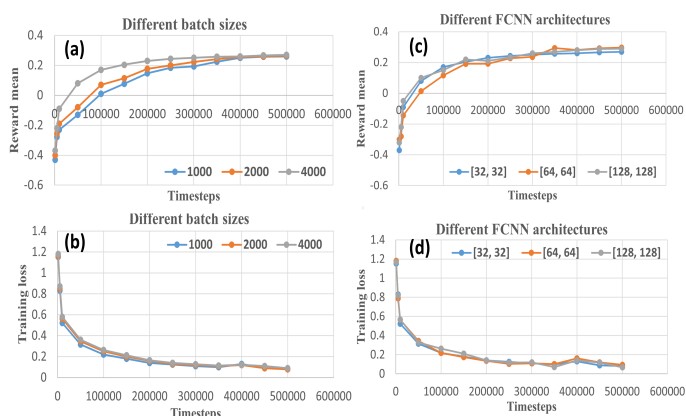

Figure 3: Reward mean and training loss as a function of the number of timesteps with different batch sizes (a-b) and FCNN architectures (c-d).

**Ablation - GNN architectures.** In this part, we investigate the effect of different GNN architectures such as GCN, GAT, and GGNN used in the supervised learning on the overall accuracy and speedup. Experimental results in Table 2 indicate that using more complicated models such as GAT and GGNN does not contribute significantly to higher accuracy and speedup compared to GCN with only 1.03x improvement. However, training GAT and GGNN takes more time compared to GCN. Therefore, we choose GCN in the current implementation.

**Reward.** We compare the DRL part in autograph and neuro-vectorizer in terms of the re-

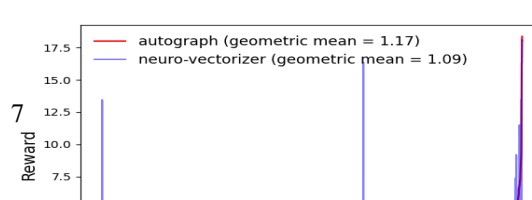

wards of all kernels in the LORE dataset. As we can see in Figure 4, many kernels have larger rewards so the red line is above the blue line. This indicates that in general, autograph could provide better performance improvement compared to neuro-vectorizer. It could be also validated by comparing the geometric mean of the rewards from autograph (1.17) and neuro-vectorizer (1.09).

**Distributions of true labels and predictions.**
We compare the distributions of true labels ranging from 0 to 24 due to 5 VFs and 5 IFs, the neuro-vectorizer predictions, and autograph predictions in LORE. As shown in Figure 5, the cumulative distribution of the autograph predictions is much closer to that of true labels than neuro-vectorizer. Furthermore, the Kullback–Leibler (KL) divergence (DIV) between autograph and true labels (1034.10 bits) is much smaller than the KL DIV between neuro-vectorizer and true labels (3995.66 bits). Therefore, it validates that autograph has a wider action distribution and is more similar to the distribution of true labels.

**Average accuracy and normalized speedup.**
We train and test all of the models on LORE. Accuracy is measured by calculating the number of times where the ground truth is the same as the predicted action, divided by the total number of kernels. The ground truth is collected from choosing the optimal execution time by exhaustively searching from the space of the vectorization parameters. The predicted action is collected from the results from each model when a new kernel arrives.

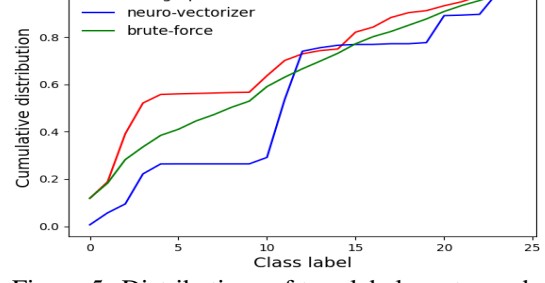

Figure 5: Distributions of true labels, autograph, and neuro-vectorizer predictions.

Tables 3, 4, and 5 show the average accuracy and speedup normalized to O3 runtimes for neuro-vectorizer, autograph, code2vec, and GCN, respectively. In NPB, Polybench, and SPEC 2006 kernels, autograph outperforms neuro-vectorizer by 5.97x, 2.2x, and 1.31x in terms of the accuracy and 1.06x, 1.16x, and 1.18x in terms of the geometric speedup compared to O3. This is because autograph learns the graph structures that can be used to predict the labels. Furthermore, compared to supervised learning, the DRL-based approaches achieve 1.22x, 1.35x, and 1.19x better in terms of speedup.

Table 3: NPB kernels.

| ML models | Original vectorization | | Effective vectorization[1] | |
|---|---|---|---|---|
| | Accuracy | Speedup | Accuracy | Speedup |
| autograph | 19.41%±1.42% | 1.16±0.05 | 20.19%±1.89% | 1.17±0.07 |
| neuro-vectorizer | 3.25%±1.88 | 1.09±0.04 | 3.80%±1.96 | 1.11±0.06 |
| code2vec | 7.72%±0.65% | 0.95±0.02 | 8.42%±0.83% | 0.97±0.03 |
| GCN | 8.91%±0.56% | 1.04±0.03 | 9.67%±0.73% | 1.07±0.05 |

[1] It is measured by pruning some kernels with less than 1.02x brute-force speedup compared to O3 baseline.

Table 4: Polybench kernels with original vectorization. Polybench is a benchmark for polyhedral transformations that operate on loops, so pruning leads to the same kernels.

| ML models | Accuracy | Speedup |
|---|---|---|
| autograph | 19.87% ± 0.53% | 1.31 ± 0.05 |
| neuro-vectorizer | 9.05% ± 1.23% | 1.13 ± 0.12 |
| code2vec | 8.22% ± 0.58% | 0.97 ± 0.04 |
| GCN | 9.80% ± 0.77% | 1.09 ± 0.08 |

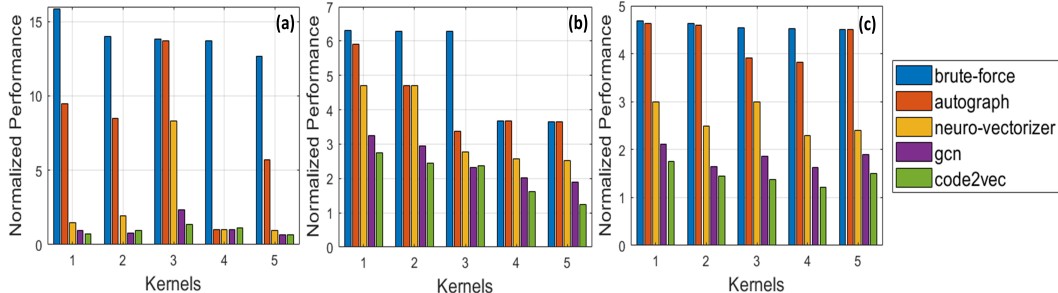

Figure 6: Performance in the SPEC 2006 (a), NPB (b), and Polybench (c) kernels with highest brute-force speedups normalized to O3.

Table 5: SPEC 2006 kernels.

| ML models | Original vectorization | | Effective vectorization | |
|---|---|---|---|---|
| | Accuracy | Speedup | Accuracy | Speedup |
| autograph | 6.42%±0.37% | 1.04±0.02 | 11.49%±0.85% | 1.24±0.04 |
| neuro-vectorizer | 4.91%±0.52% | 1.02±0.01 | 9.52%±1.08% | 1.05±0.02 |
| code2vec | 3.32%±0.22% | 0.94±0.03 | 6.19%±0.43% | 1.04±0.05 |
| GCN | 4.86%±0.41% | 0.97±0.01 | 9.15%±0.77% | 1.09±0.01 |

**Normalized performance for top kernels.** Figure 6(a-c) shows the normalized performance on SPEC 2006 (a), NPB (b), and Polybench (c) for brute-force search, supervised code2vec, supervised GCN, neuro-vectorizer, and autograph. These kernels are selected based on the top normalized speedup of brute-force compared to O3. Compared to neuro-vectorizer, autograph can provide at most 6.51x performance improvement over O3 for some files because autograph represents kernels as structured graphs and learns their representational embeddings unveiling their hidden structures.

Table 6: GCC with original vectorization. GCC is a synthetic benchmark for loop transformations, so pruning leads to the same kernels.

| ML models | Accuracy | Speedup |
|---|---|---|
| autograph | 26.51% ± 0.36% | 1.62 ± 0.07 |
| neuro-vectorizer | 9.73% ± 1.23% | 1.22 ± 0.16 |
| code2vec | 8.07% ± 0.61% | 1.08 ± 0.06 |
| GCN | 8.80% ± 0.79% | 1.15 ± 0.08 |

Table 7: MiBench subset. We use the same subset of Mibench as in neuro-vectorizer to validate the effectiveness of autograph.

| ML models | Original vectorization | | Effective vectorization | |
|---|---|---|---|---|
| | Accuracy | Speedup | Accuracy | Speedup |
| autograph | 7.25%±0.43% | 1.1±0.03 | 16.12%±0.98% | 1.41±0.07 |
| neuro-vectorizer | 5.21%±0.52% | 1.06±0.02 | 11.66%±1.23% | 1.12±0.04 |
| code2vec | 3.78%±0.24% | 0.95±0.03 | 7.76%±0.48% | 1.09±0.06 |
| GCN | 4.71%±0.45% | 1.03±0.04 | 9.75%±0.92% | 1.11±0.08 |

**Transfer learning on GCC and MiBench.** It is important to see how well the machine learning models generalize to completely new datasets. To that end, we evaluate on two benchmarks different models that are trained on LORE. GCC, compared to MiBench, is more beneficial to auto-vectorization because it is a synthetic benchmark for loops that are designed to be vectorized. As shown in Table 6 and 7, autograph can achieve on average 2.72x higher accuracy and 1.33x higher performance than neuro-vectorizer. Compared to the supervised learning models, autograph can provide on average 1.5x normalized performance improvement over baseline O3. We can conclude

that the auto-vectorization is limited to code with rich portions of loops that can be vectorized. It is not suitable for benchmarks such as MiBench where the loops constitute a small portion of the code.

## 5 CONCLUSION

We propose Autograph, an end-to-end framework for compiler auto-vectorization that automatically extracts loops, constructs dependency graphs, and learns structured representations that capture both the structural dependencies of the computation graph and the semantics of the code. Autograph uses a deep reinforcement learning approach to predict vectorization factors and injects the vectorization pragmas with the optimal VF/IF factors in the code to achieve better performance. Our extensive experiments and comparisons on multiple benchmark datasets show that for Polybench, autograph achieves on average 2.47x performance improvement for Polybench compared to neurovectorizer and 3.61x compared to the baseline.

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

## A  APPENDIX

You may include other additional sections here.

