# OpenReview forum: "Structural Code Representation Learning for Auto-Vectorization"
_ICLR.cc/2023/Conference — Submitted to ICLR 2023_

### Official Review · Reviewer_Coc4 · 2022-10-17

**Confidence:** 4
**Correctness:** 3
**Technical Novelty And Significance:** 3
**Empirical Novelty And Significance:** Not applicable
**Recommendation:** 6

**Clarity, Quality, Novelty And Reproducibility:**

## Clarity

Overall, the paper is not sufficiently clear.
Primarily, the approach is not described precisely enough for a reasonable reader to implement the approach described in the paper.
I included a number of quotations below that were not sufficiently clear to me.

* The definitions of "vectorization factor" and "interleaving factor" in Section 1 are not clear.
* "In our implementation, we use PrograML to generate code graphs.". Given that PrograML is used, what is the purpose of learning the binary classifier for link prediction
* "The forward pass allows the model to find the probability of the existence of an edge by computing a score between incident nodes with a function f". What is this function $f$?
* "Given a sequence of graphs G(V, E)..." What do these sequences of graphs represent? Is this a sequence of graphs for a specific input example, or is this sequence describing the training procedure?
* "We next learn a binary classifier ... by performing link prediction". To be clear, the binary classifier is predicting whether a link exists or not? Or does the training process use link prediction?
* "... evaluated with any metrics such as area under curve..." is AUC the evaluation metric you use?
* "...link score..." What is a link score?
* I don't understand the forall notation in equation 2.
* Equations 3 and 4 both define h_i^(l+1). Is this a typo?
* "Therefore, by adopting this approach, the size of the node embedding is the same as the size of the vocabulary". Is this ultimately the dimensionality of embedding that the GNN learns?
* "... is that two graphs could be identical but the vectorization parameters are totally different, due to different number of iterations in the loops". Maybe I misunderstand the embedding technique, but if two graphs are identical wouldn't that imply that their loop conditions are identical? If that is the case, how would there be different numbers of loop iterations?
* What are loop counts? I see a potential definition of these as "... the number of loops in each file of a benchmark", but I don't understand what this is.
* "... and further refines them in terms of better accuracy and performance improvement over O3 via graph supervised learning" what refining is this talking about?
* "...the state space consists of hidden feature embeddings learned from the supervised model". Do these states change during the execution of the agent? How?
* "The autograph framework incorporates GraphSAGE (Hamilton et al., 2017) to extract feature embeddings in an unsupervised fashion..." What are these feature embeddings used for? I don't see this described anywhere in Section 3.


## Quality

The quality of the evaluation is also not sufficient.
I am not convinced that the presented metrics (accuracy) are appropriate for evaluating the approach.
I am also not convinced by the speedup results: the train/validation/test splits aren't clear, I'm not convinced by the choice of baselines, there's no information provided on the transfer learning methodology, and the presented results for neuro-vectorizer do not align with those in the neuro-vectorizer paper.

* How is the dataset split into train/validation/test? Given that these benchmarks may share kernels, is there any deduplication performed? (Or is this not an issue)
* What is the definition of accuracy for the experimental results? Assuming it means just matching the brute-forced solution, is comparing the KL true label accuracy / divergence the best metric? This is not a binary problem -- some incorrect choices are better than others.
* How long (in terms of wall-clock time) does it take to train the model? To evaluate? To generate the brute-force solutions?
* Does autograph ever result in kernels slower than O3?
* The neuro-vectorizer paper (Figure 8) shows that neuro-vectorizer consistently achieves something close to 90% of brute force's speedup. Figure 6 in this paper shows a result closer to 50%. Why is there such a large discrepancy?

## Novelty

The paper is sufficiently novel, in that it proposes a new architecture for loop vectorization.
I do have one concern, which is that the paper itself overclaims the novelty:
* "autograph is the first to propose a structured learning framework for auto-vectorization" (Section 2): I don't think that this is true. https://dl.acm.org/doi/10.5555/3454287.3455597 also uses a graph neural network to learn a vectorization policy (albeit, using SLP rather than loop vectorization).

## Reproducibility

As discussed in the clarity section above, the paper is not sufficiently clear to be reproduced.
Beyond that, I don't see significant barriers to reproducing the results of the paper.


**Strength And Weaknesses:**

Strengths:
* The paper providers a novel approach to a relevant problem, automatic loop vectorization
* The results, if they stand up (see weaknesses) are strong, significantly outperforming O3 and prior work in neural loop vectorization

Weaknesses:
* The clarity of the approach is low (such that the approach could not reasonably be implemented by a reader) -- see the clarity section below
* The quality of the evaluation is low, such that I do not have confidence that the approach actually beats the baselines (especially the comparison to neuro-vectorizer) -- see the quality section below


**Summary Of The Paper:**

The paper presents autograph, a GNN-based approach to automatic loop vectorization.
Autograph represents the program as a graph, computes an embedding for each node using a standard GNN forward pass, then trains a RL agent to predict vectorization factors based on node embeddings.
The paper evaluates on a set of standard CPU benchmarks, showing that autograph outperforms the most related prior work in this area, neuro-vectorizer.


**Summary Of The Review:**

Reject.
The method is not sufficiently clear, such that a reasonable reader could not reimplement the approach.
The evaluation is not sufficiently high quality, such that the presented results are not convincing.

I would be willing to raise my score conditioned on:
* The authors providing a significantly more clear description of the approach.
* The authors explaining the choice of accuracy metric in the evaluation (or present a different metric)
* The authors clarifying the train/validation/test split methodology and the transfer learning methodology
* The authors explaining the discrepancy between the neuro-vectorizer results in this paper and the results in the neuro-vectorizer paper


-------

## Update after author response

Thanks to the authors for the detailed response. Based on this response, I'm raising my score to a weak accept. Specifically:
* The revision includes a much more clear description of the approach (page 4)
* The authors correctly pointed out that I missed the definition of accuracy in the original submission. Though I still think this is a weak metric, I agree with the authors that the speedup metric is ultimately the one that matters (which the paper does report).
* The author response does not state whether or not any kernels are shared between the training/test dataset (and further, do not mention any validation set). The author response is also not sufficiently clear on the response about transfer learning: my point of confusion was whether or not any fine-tuning is performed on the other task (as is typically implied by the term transfer learning). The author response implies that no fine-tuning is performed, but is not 100% clear on that.
* The explanation of the neuro-vectorizer performance is satisfactory. Though it's surprising that neuro-vectorizer does not perform anywhere near as well as in the original paper (and raises some questions about whether it is a fitting baseline as-is), I don't view this as a reason to reject this paper.

If the paper is accepted, I would still like for the authors to include the following in the final version of the paper:
* Stating to what extent the different applications in the train and test set share kernels (or if they do not share any kernels)
* Stating how hyperparameters were tuned (given the lack of a validation set, I'm assuming that everything was tuned to maximize training set accuracy, but this should be explicitly stated)
* Making clear whether any fine-tuning or other transfer learning methodology was performed, or whether the transfer learning test is just a test of out-of-distribution generalization
* Including this discussion of neuro-vectorizer results
* Further improvements to the clarity of the approach

---

> ### Author Response · Authors · 2022-11-18
> **Rebuttal**
>
> We have re-structured the graph unsupervised learning part to make it clearer. Below you can find our reponses to some of the comments due to page limit.
>
> Q: The definitions of "vectorization factor" and "interleaving factor" in Section 1 are not clear.
>
> A: As discussed in the introduction, VF determines how many instructions to pack together from different iterations of the loop. IF determines the stride of the memory accesses of the packed instructions. IF allows vectorization to be performed on non-consecutive addresses, which are generally referred to as non-unit stride accesses. Both factors are numerical numbers that are hardware-dependent to control the width of the vector instructions. The goal of the paper is to predict the two numbers based on the structural dependencies in the code.
>
> Q: How is the dataset split into train/validation/test? Transfer learning?
>
> A: As mentioned in the evaluation section, we use LORE suite of kernels to train and test DRL. The LORE benchmark suite contains multiple benchmarks: ALPBench, ASC-llnl, Cortexsuite, FreeBench, Kernels, Livermore, Mediabench, Netlib, NPB, Polybench, Scimark2, TSVC, libraries and real-world apps. Specifically, we use the SPEC 2006, NPB, and Polybench kernels as testing benhcmarks (not included during training), and the rest of benchmarks as training. The transfer learning is done after the agent in DRL is well trained with the LORE dataset. We use the well-learned agent as a predictor to transfer our knowledge of how to predict vectorization parameters  from training examples to unseen new datasets such as GCC and Mibench.
>
> Q: What is the definition of accuracy for the experimental results?
>
> A: We measure both accuracy and speedup, because as you mentioned some choices still leads to speedup over the LLVM -O3 compiler baseline, even if these choices are not the optimal. To clarify things further (see evaluation section), accuracy is measured by calculating the number of times where the ground truth is the same as the predicted label, divided by the total number of kernels, which means when the agent predicts the optimal parameter values that leads to the best execution runtime. The ground truth label is collected from choosing the parameters that lead to the optimal execution time by exhaustively searching from the space of the vectorization parameters. The predicted action is collected from the results from each model when a new kernel arrives. Either ground truth or predicted label is ranged from 0 to 24.
>
> Q: Does autograph ever result in kernels slower than O3?
>
> A: As discussed in the evaluation section, for the benchmark like SPEC 2006, the maximum speedup we can get from the brute-force solutions is only 1.06x, so both autograph and neurovectorizer could result in kernels slower than O3. Out of 1379 kernels in SPEC 2006, autograph has 18 kernels that are slower than O3. Nevertheless, autograph can still achieve 1.04x speedup compared to O3. For the benchmark like Polybench that is designed for loop transformations, autograph never results in kernels slower than O3.
>
> Q: Why is there such a large discrepancy between neurovectorizer in this paper and in the original paper?
>
> A: We use a more extensive and larger benchmark to train and test (from LORE repository, which is the state-of-the-art benchmark). In that paper, neurovectorizer is trained with the GCC benchmark and tested with Mibench and Polybench. The GCC benchmark, as mentioned in this manuscript and neurovectorizer paper, is a synthetic benchmark that contains different loops, which is not representative. Instead, in this manuscript, we use the LORE benchmark suite to train and test it. The GCC benchmark has 744 kernels whereas the LORE benchmark suite has 3928 kernels. The LORE benchmark suite contains ALPBench, ASC-llnl, Cortexsuite, FreeBench, Kernels, Livermore, Mediabench, Netlib, NPB, Polybench, Scimark2, TSVC, libraries and real-world apps. We train the agent with NPB, SPEC2006, and Polybench, and test it with the rest of the benchmarks.
>
>
> Q: I do have one concern, which is that the paper itself overclaims the novelty.
>
> A: We cited this paper but vectorization relies on two major methods: (\textit{i}) the loop vectorizer, which operates on loops, and (\textit{ii}) the superword-level parallelism (SLP) vectorizer, which operates on straight-line code. The SLP vectorizer merges multiple operations found in the code into a sequence of vectors while the loop vectorizer widens instructions in loops to operate on multiple consecutive iterations. Loops in repetitive tasks are commonly used in modern programs to save time. Therefore, in this work, we focus on the loop vectorizer, whereas the reference focuses on SLP vectorization. In the paper, we also make it more precisely by mentioning that "autograph is the first to propose a structured learning framework for loop auto-vectorization", on page 3.

---

> > ### Author Response · Authors · 2022-11-18
> > **Continued Rebuttal**
> >
> > Due to the size limitation on the rebuttal text, we respond to the rest of the questions in this comment.
> >
> > Q: "Therefore, by adopting this approach, the size of the node embedding is the same as the size of the vocabulary". Is this ultimately the dimensionality of embedding that the GNN learns?
> >
> > A: Yes, the GNN takes as input the node embeddings learned from
> > the graph unsupervised learning model. The size is the same as the size of the vocabulary.
> >
> > Q: "... is that two graphs could be identical but the vectorization parameters are totally different, due to different number of iterations in the loops". Maybe I misunderstand the embedding technique, but if two graphs are identical wouldn't that imply that their loop conditions are identical? If that is the case, how would there be different numbers of loop iterations?
> >
> > A: Imagine two kernels each containing one for loop. The number of iterations in one kernel is 8 and the number of iterations in the other kernel is 16. The structure of the programl graphs is the same, which leads to the same embedding from graph unsupervised learning. Therefore, in order to differentiate two kernels, we add the extra information in the fully connected layer in GNN to better predict the vectorization parameters.
> >
> > Q: What are loop counts? I see a potential definition of these as "... the number of loops in each file of a benchmark", but I don't understand what this is.
> >
> > A: Loop count refers to the number of iterations for any particular loop as . We changed it in the revised manuscript.
> >
> > Q: "... and further refines them in terms of better accuracy and performance improvement over O3 via graph supervised learning" what refining is this talking about?
> >
> > A. The refinement refers to the graph supervised learning that transforms the embeddings generated from graph unsupervised learning to another embeddings to provide better accuracy.
> >
> > Q: "...the state space consists of hidden feature embeddings learned from the supervised model". Do these states change during the execution of the agent? How?
> >
> > A: The embeddings do not change during the execution of the agent. The agent takes the embedding that represents a kernel as input and predicts the vectorization parameters to maximize the reward.
> >
> > Q: "The autograph framework incorporates GraphSAGE (Hamilton et al., 2017) to extract feature embeddings in an unsupervised fashion..." What are these feature embeddings used for? I don't see this described anywhere in Section 3.
> >
> > A: The feature embeddings are learned in an unsupervised way to capture the structural dependencies in the code, which is discussed at page 4 "Inductive representational graph unsupervised learning". These feature embeddings are further used in supervised learning to further refine them in terms of better accuracy.
> >
> > Q: How long (in terms of wall-clock time) does it take to train the model? To evaluate? To generate the brute-force solutions?
> >
> > A: It takes around 3-4 hours to train the model, which could be done offline. It takes around 0.2 seconds during inference to predict a label from an agent. On the other hand, brute force is an exhaustive search that is around 60000x worse compared to autograph as shown in Figure 1(b).
> >
> > Q: Does autograph ever result in kernels slower than O3?
> >
> > A: As discussed in the evaluation section, for the benchmark like SPEC 2006, the maximum speedup we can get from the brute-force solutions is only 1.06x, so both autograph and neurovectorizer could result in kernels slower than O3. Out of 1379 kernels in SPEC 2006, autograph has 18 kernels that are slower than O3. Nevertheless, autograph can still achieve 1.04x speedup compared to O3. For the benchmark like Polybench that is designed for loop transformations, autograph never results in kernels slower than O3.

---

### Official Review · Reviewer_pVcf · 2022-10-24

**Confidence:** 5
**Correctness:** 2
**Technical Novelty And Significance:** 1
**Empirical Novelty And Significance:** 2
**Recommendation:** 3

**Clarity, Quality, Novelty And Reproducibility:**

Novelty: Very low, important related work missing with no comparison. Consider [1], they use the graph structure of the program in their auto-vectorization scheme. The authors have not cited or compared against their approach, which leads to the wrong novelty claim, "In contrast to neuro-vectorizer, autograph is the first to propose a structured learning framework for auto-vectorization". I find this a major flaw in the paper and diminishes the value of the contributions.

Clarity: The unsupervised training part of the paper was confusing to me, specially its training objective. Rest of the paper was ok to read.

[1] Mendis et. al "Compiler Auto-Vectorization with Imitation Learning", NeurIPS 2019.

**Strength And Weaknesses:**

Strengths:
* The paper improves on structural representation of the code from the NeuroVectorizer paper, even though the basic premise is the same.
* Encouraging results from the LORE suite.

Weaknesses
*  Important related work that uses the structure for compiler auto-vectorization is missing. This leads to wrong novelty claims made at the end of section 2. This is a major flow.
* The paper is low in novelty, where essentially they added learning based on structure to the NeuroVectorizer. Adding structure to learning in the context of vectorization is also not new as mentioned above.

**Summary Of The Paper:**

The paper introduces a DRL based approach to perform Loop Vectorization in LLVM IR. They represent LLVM IR as a graph and then learn an embedding in an unsupervised manner using a GNN, which they later refine using a FCNN to decide the VF and IF in the loop vectorizer.

**Summary Of The Review:**

IMO, the paper lacks novelty. Important related work are missing and this leads to wrong claims. I have mentioned them under novelty. Further, important work related to cost models (e.g. [1],[2] ) and how representation learning on graphs are used to learning embeddings that can be used for this task is missing.

I find the ablations on pretraining (unsupervised learning) to be informative. However, it is unclear how the random walks are done to get negative examples. This is not the usual code language model pretraining that is used here. Better writing of this section would help. Further, it is unclear why this unsupervised learning objective was chosen, given that code language models usually opt for MLM based pertaining at least in the contextual sense.

Also, I found the "upto" speedups a little misleading considering the geomean speedups that are reported.

Overall, I think this paper is below bar for acceptance at ICLR. I am more concerned with novelty related claims.

[1] Baghdadi et al. "A Deep Learning Based Cost Model for Automatic Code Optimization" MLSys 2021

[2] Mendis et. al "Ithemal: Accurate, Portable and Fast Basic Block Throughput Estimation using Deep Neural Networks", ICML 2019

---

> ### Author Response · Authors · 2022-11-18
> **Rebuttal**
>
> Q: Important related work that uses the structure for compiler auto-vectorization is missing. This leads to wrong novelty claims made at the end of section 2. This is a major flow. The paper is low in novelty, where essentially they added learning based on structure to the NeuroVectorizer. Adding structure to learning in the context of vectorization is also not new as mentioned above. Important related work missing with no comparison. Consider [1], they use the graph structure of the program in their auto-vectorization scheme. The authors have not cited or compared against their approach, which leads to the wrong novelty claim, "In contrast to neuro-vectorizer, autograph is the first to propose a structured learning framework for auto-vectorization". I find this a major flaw in the paper and diminishes the value of the contributions.
>
> [1] Mendis et. al "Compiler Auto-Vectorization with Imitation Learning", NeurIPS 2019.
>
> A: We cited this paper on page 1-2 (Mendis et al., 2019) but vectorization relies on two major methods: (\textit{i}) the loop vectorizer, which operates on loops, and (\textit{ii}) the superword-level parallelism (SLP) vectorizer, which operates on straight-line code. The SLP vectorizer merges multiple operations found in the code into a sequence of vectors while the loop vectorizer widens instructions in loops to operate on multiple consecutive iterations. Loops in repetitive tasks are commonly used in modern programs to save time.
>
> In contrast to [1] which focuses on SLP vectorization, in this work, we focus on the loop vectorizer. None of the existing work on loop vectorization focuses on the structural dependencies in the code to predict the vectorization parameters. Therefore, [1] is not a proper baseline.
>
> Q: I find the ablations on pretraining (unsupervised learning) to be informative. However, it is unclear how the random walks are done to get negative examples. This is not the usual code language model pretraining that is used here. Better writing of this section would help.
>
> A: In the Unsupervised GraphSAGE model, node embeddings are learnt by solving a simple classification task: given a large set of “positive” (target, context) node pairs generated from random walks performed on the graph (i.e., node pairs that co-occur within a certain context window in random walks), and an equally large set of “negative” node pairs that are randomly selected from the graph according to a certain distribution, learn a binary classifier that predicts whether arbitrary node pairs are likely to co-occur in a random walk performed on the graph. Through learning this simple binary node-pair-classification task, the model automatically learns an inductive mapping from attributes of nodes and their neighbors to node embeddings in a high-dimensional vector space, which preserves structural and feature similarities of the nodes. In short, random walks are used to find node pairs in the positive node pairs. We have re-written this section in the revised manuscript.
>
>
> Q: Further, it is unclear why this unsupervised learning objective was chosen, given that code language models usually opt for MLM based pertaining at least in the contextual sense.
>
> A: The objective of unsupervised learning is to learn embeddings that better capture the structural dependencies in the code, that can be used later in the graph supervised learning.
>
> Q: Also, I found the "upto" speedups a little misleading considering the geomean speedups that are reported.
>
> A: Thanks, we fixed this and provide the average instead in the paper.

---

### Official Review · Reviewer_NSgq · 2022-10-25

**Confidence:** 4
**Clarity, Quality, Novelty And Reproducibility:** The writing is clear. The idea is new…
**Correctness:** 3
**Technical Novelty And Significance:** 3
**Empirical Novelty And Significance:** 2
**Recommendation:** 5

**Strength And Weaknesses:**

## Strength
The paper is well writing. The presentation of the technique is clear.

## Weakness
1. While the presentation of the technique is clear, it is unclear why it outperforms previous method. Can you give an example that shows your method obtains better vectorization than previous methods?

2. I do not find any source code. To make your work more convincing, please consider open source your code. Auto-vectorization is a well studied topic in compiler community and many techniques are implemented in real compilers. If you want to claim an advantage over the existing work, you need to show the code.

**Summary Of The Paper:**

This paper presents a learning-based auto-vectorization method. The idea is to represent the code as a graph, learning embeddings for nodes in the graph, and use the embeddings to predict two vectorization parameters: VF and IF.

**Summary Of The Review:**

The paper is easy to follow. The techniques look new to me. However, for such learning-based code optimization work, a real implementation is important. For this reason, I give the paper a weak reject.

---

> ### Author Response · Authors · 2022-11-18
> **Rebuttal**
>
> Q: While the presentation of the technique is clear, it is unclear why it outperforms previous method. Can you give an example that shows your method obtains better vectorization than previous methods?
>
> A: As mentioned in the related work, compared to previous methods such as neuro-vectorizer, autograph is the first to propose a structured learning framework for auto-vectorization which leverages code dependencies that are not captured by neuro-vectorizer. More precisely, our approach  converts code to graphs and learns structured representations from code. This structured representation can better capture the dependencies among instructions compared to previous work that only relies on the code text features. This can best be observed from the results shown in Tables 3, 4 and 5, as well as Figures 6. For example, autograph can provide kernel 3 in SPEC 2006 with the speedup that is very close to the brute-force method and offer 1.65x performance improvement over the state-of-the-art technique neurovectorizer and 13.78x over the baseline LLVM O3 because autograph takes into account the structural dependencies in the code to predict vectorization parameters.
>
> The proposed approach, namely the autograph, can be used across different architectures with different SIMD capabilities. For example, if a new kernel is developed or needs to be considered in a complex autonomous system, we first analyze the dependencies among data, control, and function calls to construct a graph representation of this kernel code. The graph is then transformed into an embedding via graph unsupervised learning, which is further refined to optimize the accuracy via graph supervised learning. The learned embedding are used as an input to the learned agent in deep reinforcement learning to predict VF and IF, two vectorization parameters.
>
> Q: I do not find any source code. To make your work more convincing, please consider open source your code. Auto-vectorization is a well studied topic in compiler community and many techniques are implemented in real compilers. If you want to claim an advantage over the existing work, you need to show the code.
>
> A: We provided the code at this anonymized Github link: \href{https://anonymous.4open.science/r/autograph-029E/README} {autograph}
> We have uploaded the code to Github but since the review is required to be anonymous, we didn't provide the link initially, we apologize for this. However, we are happy to provide the source code for the framework that can be accessed from the following link: \href{https://anonymous.4open.science/r/autograph-029E/README} {autograph} and it can be verified by anyone interested for accuracy and reproducibility purposes.

---

### Official Review · Reviewer_9X1Y · 2022-11-03

**Confidence:** 4
**Correctness:** 2
**Technical Novelty And Significance:** 2
**Empirical Novelty And Significance:** 2
**Recommendation:** 3

**Clarity, Quality, Novelty And Reproducibility:**

The writing of this paper needs further improvement. Important baselines are missing in the experiment. The rationale of employing deep reinforcement learning in the proposed method remains unclear.

**Strength And Weaknesses:**

There are some weaknesses of this paper:

-	The rationale of employing deep reinforcement learning in the proposed method remains unclear. More discussion and empirical evidence (e.g., the ablation study) should be provided about how the deep RL helps the loop-based vectorization problem.
-	Important baselines are missing in the experiment. The state-of-the-art loop-based vectorization method (e.g., Ref Wang and O’Boyle, 2018) is missing in the comparison, which brings more concerns about the performance improvement of the proposed method.
-	In addition, the loop-based vectorization problem is likely to have an impact within only a subfield of learning representation, which reduces the significance of this paper.


**Summary Of The Paper:**

This paper mainly focuses on the problem of loop-based vectorization and proposes a graph-based learning framework named autograph. Autograph aims to automatically predict the correct vectorization factor and the interleaving factor. Experimental results based on several data sets are reported.

**Summary Of The Review:**

The rationale of employing deep reinforcement learning in the proposed method remains unclear. Important baselines are missing in the experiment. In addition, the loop-based vectorization problem is likely to have an impact within only a subfield of learning representation.

---

> ### Author Response · Authors · 2022-11-16
> **Rebuttal**
>
> Q: The rationale of employing deep reinforcement learning in the proposed method remains unclear. More discussion and empirical evidence (e.g., the ablation study) should be provided about how the deep RL helps the loop-based vectorization problem.
>
> A: As discussed in the original version of the manuscript, the rationale of employing the proposed end-to-end RL framework is that (1) it does not require labels to train the agent to prevent the time-consuming exhaustive search; (2) compared to supervised learning that optimizes the accuracy, it relies on the reward function that involves the normalized execution time to improve the system-level speedup; (3) it is hardware-agnostic, meaning that while switching to another hardware platform, this RL framework does not require the exhaustive search to collect new labels before training compared to supervised learning.
>
> The ablation study was provided in the evaluation section on page 7 where we evaluate each part of the framework to understand its contributions on the speedup and several graph neural network architectures such as graph convolutional networks, graph attention network, and gated graph neural networks.
>
> Q: Important baselines are missing in the experiment. The state-of-the-art loop-based vectorization method (e.g., Ref Wang and O’Boyle, 2018) is missing in the comparison, which brings more concerns about the performance improvement of the proposed method.
>
> A: We cite the reference by Wang and O'Boyle on page 3 in Section 2. However, this reference is not a proper baseline for autovectorization because it only provides a general view (survey) of how to do feature engineering for using machine learning for compiler optimization. In addition, the reference by Wang and O'Boyle does not study the autovectorization problem and does not provide a solution strategy for our problem that we can compare against.
>
> Most importantly, in contrast to state-of-the-art, we provide a fully data-driven approach that exploits the structural dependencies in the code and predict the vectorization parameters using deep reinforcement learning without relying on hand-engineered features (as in Wang and O'Boyle). As such, we compare our framework against the state-of-the-art technique neuro-vectorizer (i.e., ref [1] below).
>
> [1] Haj-Ali, A., Ahmed, N. K., Willke, T., Shao, Y. S., Asanovic, K., \& Stoica, I. (2020, February). Neurovectorizer: End-to-end vectorization with deep reinforcement learning. In Proceedings of the 18th ACM/IEEE International Symposium on Code Generation and Optimization (pp. 242-255).

---

### Decision · Program_Chairs · 2023-01-20

**Decision:**

Reject

**Justification For Why Not Higher Score:**

Unresolved questions about originality with respect to the representation learning problems being tackled.

**Justification For Why Not Lower Score:**

N/A

**Metareview: Summary, Strengths And Weaknesses:**

This paper proposes a method for predicting parameters of compiler optimizations related to vectorization of loops. It leverages graph representations of code and deep reinforcement learning. The main debate in the reviews and rebuttals is around novelty. Reviewers point out that the problem setup and deep RL formulation are shared with NeuroVectorizer, and that adding structure to learning in the context of vectorization has been done by Mendis et. al (2019). Thus the perceived originality is low. Authors respond by saying that Mendis et al tackle a different variant of vectorization, so it is not a proper baseline, and they, e.g., argue for the importance of their chosen in the response to Coc4 by saying that it's commonly occurring in practice. Unfortunately, IMO, this is missing the point. The question isn't whether this is the first paper to tackle this particular variant in this particular way. The question is for the ICLR audience, why the distinction matters. I wish that the paper (and rebuttals) would spend more time on what unique representation learning challenges arise in their setting, and what new representation learning insights arise based on their experiments, compared to the baselines being discussed.